# Time-varying Graph Representation Learning via Higher-Order Skip-Gram with Negative Sampling

## Abstract

Representation learning models for graphs are a successful family of techniques that project nodes into feature spaces that can be exploited by other machine learning algorithms. Since many real-world networks are inherently dynamic, with interactions among nodes changing over time, these techniques can be defined both for static and for time-varying graphs. Here, we show how the skip-gram embedding approach can be used to perform implicit tensor factorization on different tensor representations of time-varying graphs. We show that higher-order skip-gram with negative sampling (HOSGNS) is able to disentangle the role of nodes and time, with a small fraction of the number of parameters needed by other approaches. We empirically evaluate our approach using time-resolved face-to-face proximity data, showing that the learned representations outperform state-of-the-art methods when used to solve downstream tasks such as network reconstruction. Good performance on predicting the outcome of dynamical processes such as disease spreading shows the potential of this new method to estimate contagion risk, providing early risk awareness based on contact tracing data. The source code and data are publicly available at [link to anonymized repository].

## 1 Introduction

A great variety of natural and artificial systems can be represented as networks of elementary structural entities coupled by relations between them. The abstraction of such systems as networks helps us understand, predict and optimize their behaviour (Newman, 2003; Albert & Barabási, 2002). In this sense, node and graph embeddings have been established as standard feature representations in many learning tasks (Cai et al., 2018; Goyal & Ferrara, 2018). Node embedding methods map nodes into low-dimensional vectors that can be used to solve downstream tasks such as edge prediction, network reconstruction and node classification.

Node embeddings have proven successful in achieving low-dimensional encoding of static network structures, but many real-world networks are inherently dynamic (Holme & Saramäki, 2012). Time-resolved networks are also the support of important dynamical processes, such as epidemic or rumor spreading, cascading failures, consensus formation, etc. (Barrat et al., 2008). Time-resolved node embeddings have been shown to yield improved performance for predicting the outcome of dynamical processes over networks, such as information diffusion and disease spreading (Sato et al., 2019), providing estimation of infection and contagion risk when used with contact tracing data.

Since we expect having more data on proximity networks being used for contact tracing and as proxies for epidemic risk (Alsdurf et al., 2020), learning meaningful representations of time-resolved proximity networks can be of extreme importance when facing events such as epidemic outbreaks (Kapoor et al., 2020; Gao et al., 2020). The manual and automatic collection of time-resolved proximity graphs for contact tracing purposes presents an opportunity for quick identification of possible infection clusters and infection chains. Even before the COVID-19 pandemic, the use of wearable proximity sensors for collecting time-resolved proximity networks has been largely discussed in the literature and many approaches have been used to describe patterns of activity and community structure, and to study spreading patterns of infectious diseases (Sapienza et al., 2015; Gauvin et al., 2014; Génois et al., 2015).

Here we propose a representation learning model that performs implicit tensor factorization on different higher-order representations of time-varying graphs. The main contributions are as follows: Given that the skip-gram embedding approach implicitly performs a factorization of the shifted *pointwise mutual information* matrix (PMI) (Levy & Goldberg, 2014), we generalize it to perform implicit factorization of a shifted PMI tensor. We then define the steps to achieve this factorization using higher-order skip-gram with negative sampling (HOSGNS) optimization. We show how to apply 3rd-order and 4th-order SGNS on different higher-order representations of time-varying graphs. Finally, we show that time-varying graph representations learned via HOSGNS outperform state-of-the-art methods when used to solve downstream tasks, even using a fraction of the number of embedding parameters.

We report the results of learning embeddings on empirical time-resolved face-to-face proximity data and using such representations as predictors for solving two different tasks: network reconstruction and predicting the outcomes of a SIR spreading process over the time-varying graph. We compare these results with state-of-the art methods for time-varying graph representation learning.

## 2 PRELIMINARIES AND RELATED WORK

**Skip-gram representation learning.** The skip-gram model was designed to compute word embeddings in WORD2VEC (Mikolov et al., 2013), and afterwards extended to graph node embeddings (Perozzi et al., 2014; Tang et al., 2015; Grover & Leskovec, 2016). Levy & Goldberg (2014) established the relation between skip-gram trained with negative sampling (SGNS) and traditional low-rank approximation methods (Kolda & Bader, 2009; Anandkumar et al., 2014), showing the equivalence of SGNS optimization to factorizing a shifted PMI matrix (Church & Hanks, 1990). This equivalence was later retrieved from diverse assumptions (Assylbekov & Takhanov, 2019; Allen et al., 2019; Melamud & Goldberger, 2017; Arora et al., 2016; Li et al., 2015), and exploited to compute closed form expressions approximated in different graph embedding models (Qiu et al., 2018). In this work, we refer to the shifted PMI matrix also as $\text{SPMI}_\kappa = \text{PMI} - \log \kappa$, where $\kappa$ is the number of negative samples.

**Random walk based graph embeddings.** Given an undirected, weighted and connected graph $\mathcal{G} = (\mathcal{V}, \mathcal{E})$ with nodes $i, j \in \mathcal{V}$, edges $(i, j) \in \mathcal{E}$ and adjacency matrix $\mathbf{A}$, graph embedding methods are unsupervised models designed to map nodes into dense $d$-dimensional representations $(d \ll |\mathcal{V}|)$ (Hamilton et al., 2017). A well known family of approaches based on the skip-gram model consists in sampling random walks from the graph and processing node sequences as textual sentences. In DEEPWALK (Perozzi et al., 2014) and NODE2VEC (Grover & Leskovec, 2016), the skip-gram model is used to obtain node embeddings from co-occurrences in random walk realizations. Although the original implementation of DEEPWALK uses hierarchical softmax to compute embeddings, we will refer to the SGNS formulation given by Qiu et al. (2018).

Since SGNS can be interpreted as a factorization of the word-context PMI matrix (Levy & Goldberg, 2014), the asymptotic form of the PMI matrix implicitly decomposed in DEEPWALK can be derived (Qiu et al., 2018). Given the 1-step transition matrix $\mathbf{P} = \mathbf{D}^{-1}\mathbf{A}$, where $\mathbf{D} = \text{diag}(d_1, \ldots, d_{|\mathcal{V}|})$ and $d_i = \sum_{j \in \mathcal{V}} \mathbf{A}_{ij}$ is the (weighted) node degree, the expected PMI for a node-context pair $(i, j)$ occurring in a $T$-sized window is:

$$\mathbb{E}[\,\text{PMI}^{\text{DEEPWALK}}(i, j) \mid T\,] = \log \left( \frac{\frac{1}{2T}\sum_{r=1}^{T}\left[p^*(i)(\mathbf{P}^r)_{ij} + p^*(j)(\mathbf{P}^r)_{ji}\right]}{p^*(i)\,p^*(j)} \right) \tag{2.1}$$

where $p^*(i) = \frac{d_i}{\text{vol}(\mathcal{G})}$ is the unique stationary distribution for random walks (Masuda et al., 2017) and $\text{vol}(\mathcal{G}) = \sum_{i,j \in \mathcal{V}} \mathbf{A}_{ij}$. We will use this expression in Section 3.2 to build PMI tensors from higher-order graph representations.

**Time-varying graphs and their algebraic representations.** Time-varying graphs (Holme & Saramäki, 2012) are defined as triples $\mathcal{H} = (\mathcal{V}, \mathcal{E}, \mathcal{T})$, i.e. collections of events $(i, j, k) \in \mathcal{E}$, representing undirected pairwise relations among nodes at discrete times $(i, j \in \mathcal{V}, k \in \mathcal{T})$. $\mathcal{H}$ can be seen as a temporal sequence of static graphs $\{\mathcal{G}^{(k)}\}_{k \in \mathcal{T}}$, each of those with adjacency matrix $\mathbf{A}^{(k)}$ such that $\mathbf{A}_{ij}^{(k)} = \omega(i, j, k) \in \mathbb{R}$ is the weight of the event $(i, j, k) \in \mathcal{E}$. We can concatenate the list of time-stamped snapshots $[\mathbf{A}^{(1)}, \ldots, \mathbf{A}^{(|\mathcal{T}|)}]$ to obtain a single 3rd-order tensor

$\mathcal{A}^{stat}(\mathcal{H}) \in \mathbb{R}^{|\mathcal{V}| \times |\mathcal{V}| \times |\mathcal{T}|}$ which characterize the evolution of the graph over time. This representation has been used to discover latent community structures of temporal graphs (Gauvin et al., 2014) and to perform temporal link prediction (Dunlavy et al., 2011). Indeed, beyond the above stacked graph representation, more exhaustive representations are possible. In particular, the multi-layer approach (De Domenico et al., 2013) allows to map the topology of a time-varying graph $\mathcal{H}$ into a static network $\mathcal{G}_{\mathcal{H}} = (\mathcal{V}_{\mathcal{H}}, \mathcal{E}_{\mathcal{H}})$ (the *supra-adjacency* graph) such that vertices of $\mathcal{G}_{\mathcal{H}}$ correspond to pairs $(i, k) \equiv i^{(k)} \in \mathcal{V} \times \mathcal{T}$ of the original time-dependent network. This representation can be stored in a 4th-order tensor $\mathcal{A}^{dyn}(\mathcal{H}) \in \mathbb{R}^{|\mathcal{V}| \times |\mathcal{V}| \times |\mathcal{T}| \times |\mathcal{T}|}$ equivalent, up to an opportune reshaping, to the adjacency matrix $\mathbf{A}(\mathcal{G}_{\mathcal{H}}) \in \mathbb{R}^{|\mathcal{V}||\mathcal{T}| \times |\mathcal{V}||\mathcal{T}|}$ associated to $\mathcal{G}_{\mathcal{H}}$. Multi-layer representations for time-varying networks have been used to study time-dependent centrality measures (Taylor et al., 2019) and properties of spreading processes (Valdano et al., 2015).

**Time-varying graph representation learning.** Given a time-varying graph $\mathcal{H} = (\mathcal{V}, \mathcal{E}, \mathcal{T})$, we define as temporal network embedding a model that learns from data, implicitly or explicitly, a mapping function:

$$f : (v, t) \in \mathcal{V} \times \mathcal{T} \mapsto \mathbf{v}^{(t)} \in \mathbb{R}^d \tag{2.2}$$

which project time-stamped nodes into a latent low-rank vector space that encodes structural and temporal properties of the original evolving graph. Many existing methods learn node representations from sequences of static snapshots through incremental updates in a streaming scenario: deep autoencoders (Goyal et al., 2017), SVD (Zhang et al., 2018), skip-gram (Du et al., 2018) and random walk sampling (Béres et al., 2019; Mahdavi et al., 2018; Yu et al., 2018). Another class of models learn dynamic node representations by recurrent/attention mechanisms (Goyal et al., 2020; Li et al., 2018; Sankar et al., 2020; Xu et al., 2020) or by imposing temporal stability among adjacent time intervals (Zhou et al., 2018; Zhu et al., 2016). DYANE (Sato et al., 2019) and WEG2VEC (Torricelli et al., 2020) project the dynamic graph structure into a static graph, in order to compute embeddings with WORD2VEC. Closely related to these ones are Nguyen et al. (2018) and Zhan et al. (2020), which learn node vectors according to time-respecting random walks or spreading trajectory paths. Moreover, Kumar et al. (2019) proposed an embedding framework for user-item temporal interactions, and Malik et al. (2020) suggested a tensor-based convolutional architecture for dynamic graphs.

Methods that perform well for predicting outcomes of spreading processes make use of time-respecting supra-adjacency representations such as the one proposed by Valdano et al. (2015). In this representation, random itineraries correspond to temporal paths of the original time-varying graph. The supra-adjacency representation $\mathcal{G}_{\mathcal{H}}$ that we refer in Section 3.2, also used in DYANE, with adjacency matrix $\mathbf{A}(\mathcal{G}_{\mathcal{H}})$, is defined by two rules:

1. For each event $(i, j, t_0)$, if $i$ is also active at time $t_1 > t_0$ and in no other time-stamp between the two, we add a *cross-coupling* edge between supra-adjacency nodes $j^{(t_0)}$ and $i^{(t_1)}$. In addition, if the next interaction of $j$ with other nodes happens at $t_2 > t_0$, we add an edge between $i^{(t_0)}$ and $j^{(t_2)}$. The weights of such edges are set to $\omega(i, j, t_0)$.

2. For every case as described above, we also add *self-coupling* edges $(i^{(t_0)}, i^{(t_1)})$ and $(j^{(t_0)}, j^{(t_2)})$, with weights set to 1.

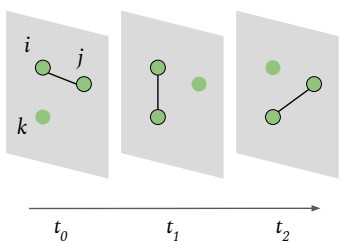 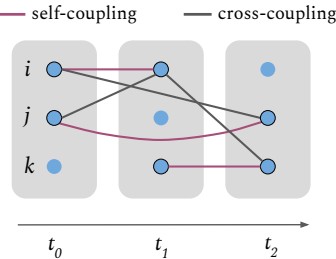

Figure 1: A time-varying graph $\mathcal{H}$ with three intervals (left) and its corresponding time-respecting supra-adjacency graph $\mathcal{G}_{\mathcal{H}}$ (right).

Figure 1 shows the differences between a time-varying graph and its time-aware supra-adjacency representation, according to the formulation described above. DYANE computes, given a node $i \in \mathcal{V}$, one vector representation for each time-stamped node $i^{(t)} \in \mathcal{V}^{(\mathcal{T})} = \{(i, t) \in \mathcal{V} \times \mathcal{T} : \exists\, (i, j, t) \in \mathcal{E}\}$ of this supra-adjacency representation. Similar models that learn time-resolved node representations

require a quantity $\mathcal{O}(|\mathcal{V}| \cdot |\mathcal{T}|)$ of embedding parameters to represent the system in the latent space. Compared with these methods, our approach requires a quantity $\mathcal{O}(|\mathcal{V}| + |\mathcal{T}|)$ of embedding parameters for disentangled node and time representations.

## 3 PROPOSED METHOD

Given a time-varying graph $\mathcal{H} = (\mathcal{V}, \mathcal{E}, \mathcal{T})$, we propose a representation learning method that learns disentangled representations for nodes and time slices. More formally, we learn a function:

$$f^* : (v, t) \in \mathcal{V} \times \mathcal{T} \mapsto \mathbf{v}, \mathbf{t} \in \mathbb{R}^d$$

through a number of parameters proportional to $\mathcal{O}(|\mathcal{V}| + |\mathcal{T}|)$. This embedding representation can then be reconciled with the definition in Eq. (2.2) by combining $\mathbf{v}$ and $\mathbf{t}$ in a single $\mathbf{v}^{(t)}$ representation using any combination function $c : (\mathbf{v}, \mathbf{t}) \in \mathbb{R}^d \times \mathbb{R}^d \mapsto \mathbf{v}^{(t)} \in \mathbb{R}^d$.

Starting from the existing skip-gram framework for node embeddings, we propose a higher-order generalization of skip-gram with negative sampling (HOSGNS) applied to time-varying graphs. We show that it allows to implicitly factorize higher-order relations that characterize tensor representations of time-varying graphs, in the same way that the classical SGNS decomposes dyadic relations associated to a static graph. Similar approaches have been applied in NLP for dynamic word embeddings (Rudolph & Blei, 2018), and higher-order extensions of the skip-gram model have been proposed to learn context-dependent (Liu et al., 2015) and syntactic-aware (Cotterell et al., 2017) word representations. Moreover tensor factorization techniques have been applied to include the temporal dimension in recommender systems (Xiong et al., 2010; Wu et al., 2019), knowledge graphs (Lacroix et al., 2020; Ma et al., 2019) and face-to-face contact networks (Sapienza et al., 2015; Gauvin et al., 2014). But this work is the first to merge SGNS with tensor factorization, and then apply it to learn time-varying graph embeddings.

### 3.1 HIGHER-ORDER SKIP-GRAM WITH NEGATIVE SAMPLING AS IMPLICIT TENSOR FACTORIZATION

Here we address the problem of generalizing SGNS to learn embedding representations from higher-order co-occurrences. We analyze here the 3rd-order case, giving the description of the general $N$-order case in the Supplementary Information. Later in this work we will focus 3rd and 4th order representations since these are the most interesting for time-varying graphs.

We consider a set of training samples $\mathcal{D} = \{(i, j, k), \ i \in \mathcal{W}, \ j \in \mathcal{C}, \ k \in \mathcal{T}\}$ obtained by collecting co-occurrences among elements from three sets $\mathcal{W}$, $\mathcal{C}$ and $\mathcal{T}$. While SGNS is limited to pairs of node-context $(i, j)$, here $\mathcal{D}$ is constructed with three (or more) variables, e.g. sampling random walks over a higher-order data structure. We denote as $\#(i, j, k)$ the number of times the triple $(i, j, k)$ appears in $\mathcal{D}$. Similarly we use $\#i = \sum_{j,k} \#(i, j, k)$, $\#j = \sum_{i,k} \#(i, j, k)$ and $\#k = \sum_{i,j} \#(i, j, k)$ as the number of times each distinct element occurs in $\mathcal{D}$, with relative frequencies $P_\mathcal{D}(i, j, k) = \frac{\#(i,j,k)}{|\mathcal{D}|}$, $P_\mathcal{D}(i) = \frac{\#i}{|\mathcal{D}|}$, $P_\mathcal{D}(j) = \frac{\#j}{|\mathcal{D}|}$ and $P_\mathcal{D}(k) = \frac{\#k}{|\mathcal{D}|}$.

Optimization is performed as a binary classification task, where the objective is to discern occurrences actually coming from $\mathcal{D}$ from random occurrences. We define the likelihood for a single observation $(i, j, k)$ by applying a sigmoid ($\sigma(x) = (1 + e^{-x})^{-1}$) to the higher-order inner product $[\![\cdot]\!]$ of corresponding $d$-dimensional representations:

$$P[\,(i, j, k) \in \mathcal{D} \mid \mathbf{w}_i, \mathbf{c}_j, \mathbf{t}_k\,] = \sigma\big(\,[\![\mathbf{w}_i, \mathbf{c}_j, \mathbf{t}_k]\!]\,\big) \equiv \sigma\left(\sum_{r=1}^d \mathbf{W}_{ir} \mathbf{C}_{jr} \mathbf{T}_{kr}\right) \tag{3.1}$$

where embedding vectors $\mathbf{w}_i, \mathbf{c}_j, \mathbf{t}_k \in \mathbb{R}^d$ are respectively rows of $\mathbf{W} \in \mathbb{R}^{|\mathcal{W}| \times d}$, $\mathbf{C} \in \mathbb{R}^{|\mathcal{C}| \times d}$ and $\mathbf{T} \in \mathbb{R}^{|\mathcal{T}| \times d}$. In the 4th-order case we will also have a fourth embedding matrix $\mathbf{S} \in \mathbb{R}^{|\mathcal{S}| \times d}$ related to a fourth set $\mathcal{S}$. For negative sampling we fix an observed $(i, j, k) \in \mathcal{D}$ and independently sample $j_\mathcal{N}$ and $k_\mathcal{N}$ to generate $\kappa$ negative examples $(i, j_\mathcal{N}, k_\mathcal{N})$. In this way, for a single occurrence $(i, j, k) \in \mathcal{D}$, the expected contribution to the loss is:

$$\ell(i, j, k) = \log \sigma\big([\![\mathbf{w}_i, \mathbf{c}_j, \mathbf{t}_k]\!]\big) + \kappa \cdot \mathop{\mathbb{E}}_{j_\mathcal{N}, k_\mathcal{N} \sim P_\mathcal{N}}\Big[\log \sigma\big(-[\![\mathbf{w}_i, \mathbf{c}_{j_\mathcal{N}}, \mathbf{t}_{k_\mathcal{N}}]\!]\big)\Big] \tag{3.2}$$

where the noise distribution is the product of independent marginal probabilities $P_\mathcal{N}(j,k) = P_\mathcal{D}(j) \cdot P_\mathcal{D}(k)$. Thus the global objective is the sum of all the quantities of Eq. (3.2) weighted with the corresponding relative frequency $P_\mathcal{D}(i,j,k)$. The full loss function can be expressed as:

$$\mathcal{L} = -\sum_{i=1}^{|\mathcal{W}|}\sum_{j=1}^{|\mathcal{C}|}\sum_{k=1}^{|\mathcal{T}|} \Big[ P_\mathcal{D}(i,j,k)\log\sigma\big(\llbracket \mathbf{w}_i, \mathbf{c}_j, \mathbf{t}_k \rrbracket \big) + \kappa\, P_\mathcal{N}(i,j,k)\log\sigma\big(-\llbracket \mathbf{w}_i, \mathbf{c}_j, \mathbf{t}_k \rrbracket \big) \Big] \quad (3.3)$$

In Supplementary Information we show the formal steps to obtain Eq. (3.3) for the $N$-order case and that it can be optimized with respect to the embedding parameters, satisfying the low-rank tensor approximation of the multivariate shifted PMI tensor into factor matrices $\mathbf{W}, \mathbf{C}, \mathbf{T}$:

$$\sum_{r=1}^{d} \mathbf{W}_{ir}\mathbf{C}_{jr}\mathbf{T}_{kr} \approx \log\left(\frac{P_\mathcal{D}(i,j,k)}{P_\mathcal{N}(i,j,k)}\right) - \log\kappa \equiv \mathrm{SPMI}_\kappa(i,j,k) \quad (3.4)$$

## 3.2 Time-varying graph embedding via HOSGNS

While a static graph $\mathcal{G} = (\mathcal{V}, \mathcal{E})$ is uniquely represented by an adjacency matrix $\mathbf{A}(\mathcal{G}) \in \mathbb{R}^{|\mathcal{V}|\times|\mathcal{V}|}$, a time-varying graph $\mathcal{H} = (\mathcal{V}, \mathcal{E}, \mathcal{T})$ admits diverse possible higher-order adjacency relations (Section 2). Starting from these higher-order relations, we can either use them directly or use random walk realizations to build a dataset of higher-order co-occurrences. In the same spirit that random walk realizations give place to dyadic co-occurrences used to learn embeddings in SGNS, we use higher-order co-occurrences to learn embeddings via HOSGNS.

As discussed in Section 3.1, the statistics of higher-order relations can be summarized in multivariate PMI tensors, which derive from proper co-occurrence probabilities among elements. Once such PMI tensors are constructed, we can again factorize them via HOSGNS. To show the versatility of this approach, we choose PMI tensors derived from two different types of higher-order relations:

1. A 3rd-order tensor $\boldsymbol{\mathcal{P}}^{(stat)}(\mathcal{H}) \in \mathbb{R}^{|\mathcal{V}|\times|\mathcal{V}|\times|\mathcal{T}|}$ which gather relative frequencies of nodes occurrences in temporal edges:

$$(\boldsymbol{\mathcal{P}}^{(stat)})_{ijk} = \frac{\omega(i,j,k)}{\mathrm{vol}(\mathcal{H})} \quad (3.5)$$

   where $\mathrm{vol}(\mathcal{H}) = \sum_{i,j,k}\omega(i,j,k)$ is the total weight of interactions occurring in $\mathcal{H}$. These probabilities are associated to the snapshot sequence representation $\boldsymbol{\mathcal{A}}^{stat}(\mathcal{H}) = [\mathbf{A}^{(1)}, \dots, \mathbf{A}^{(|\mathcal{T}|)}]$ and contain information about the topological structure of $\mathcal{H}$.

2. A 4th-order tensor $\boldsymbol{\mathcal{P}}^{(dyn)}(\mathcal{H}) \in \mathbb{R}^{|\mathcal{V}|\times|\mathcal{V}|\times|\mathcal{T}|\times|\mathcal{T}|}$, which gather occurrence probabilities of time-stamped nodes over random walks of the supra-adjacency graph $\mathcal{G}_\mathcal{H}$ (as used in DyANE). Using the numerator of Eq. (2.1) tensor entries are given by:

$$(\boldsymbol{\mathcal{P}}^{(dyn)})_{ijkl} = \frac{1}{2T}\sum_{r=1}^{T}\left[\frac{d_{(ik)}}{\mathrm{vol}(\mathcal{G}_\mathcal{H})}(\mathbf{P}^r)_{(ik)(jl)} + \frac{d_{(jl)}}{\mathrm{vol}(\mathcal{G}_\mathcal{H})}(\mathbf{P}^r)_{(jl)(ik)}\right] \quad (3.6)$$

   where $(ik)$ and $(jl)$ are lexicographic indices of the supra-adjacency matrix $\mathbf{A}(\mathcal{G}_\mathcal{H})$ corresponding to nodes $i^{(k)}$ and node $j^{(l)}$. These probabilities encode causal dependencies among temporal nodes and are correlated with dynamical properties of spreading processes.

We also combined the two representations in a single tensor that is the average of $\boldsymbol{\mathcal{P}}^{(stat)}$ and $\boldsymbol{\mathcal{P}}^{(dyn)}$

$$(\boldsymbol{\mathcal{P}}^{(stat|dyn)})_{ijkl} = \frac{1}{2}\left[(\boldsymbol{\mathcal{P}}^{(stat)})_{ijk}\delta_{kl} + (\boldsymbol{\mathcal{P}}^{(dyn)})_{ijkl}\right] \quad (3.7)$$

where $\delta_{kl} = \mathbb{1}[k=l]$ is the Kronecker delta.

Figure 2 summarizes the differences between graph embedding via classical SGNS and time-varying graph embedding via HOSGNS. Here, indices $(i,j,k,l)$ correspond to *(node, context, time, context-time)* in a 4th-order tensor representation of $\mathcal{H}$.

The above tensors gather empirical probabilities $P_\mathcal{D}(i,j,k\dots)$ corresponding to positive examples of observable higher-order relations. The probabilities of negative examples $P_\mathcal{N}(i,j,k\dots)$ can be

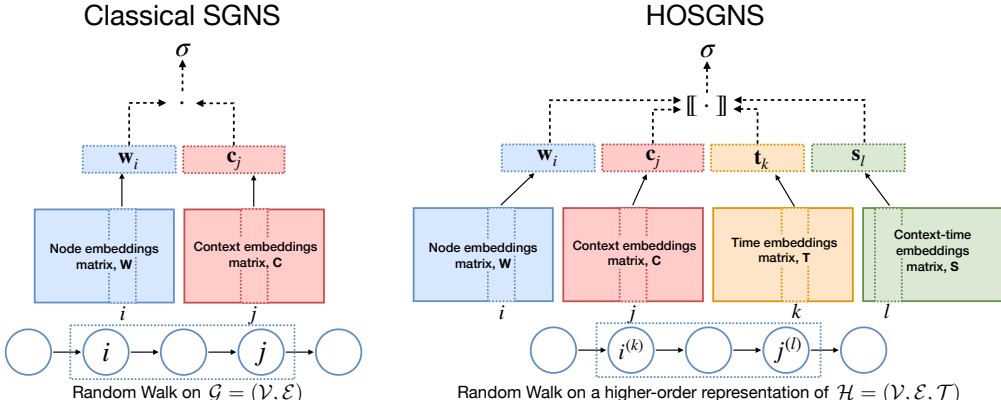

Figure 2: Representation of SGNS and HOSGNS with embedding matrices and operations on embedding vectors. Starting from a random walk realization on a static graph $\mathcal{G} = (\mathcal{V}, \mathcal{E})$, SGNS takes as input nodes $i$ and $j$ within a context window of size $T$, and maximizes $\sigma(\mathbf{w}_i \cdot \mathbf{c}_j)$. HOSGNS starts from a random walk realization on a higher-order representation of time-varying graph $\mathcal{H} = (\mathcal{V}, \mathcal{E}, \mathcal{T})$, takes as input nodes $i^{(k)}$ (node $i$ at time $k$) and $j^{(l)}$ (node $j$ at time $l$) within a context window of size $T$ and maximizes $\sigma(\llbracket \mathbf{w}_i, \mathbf{c}_j, \mathbf{t}_k, \mathbf{s}_l \rrbracket)$. In both cases, for each input sample, we fix $i$ and draw $\kappa$ combinations of $j$ or $j, k, l$ from a noise distribution, and we maximize $\sigma(-\mathbf{w}_i \cdot \mathbf{c}_j)$ (SGNS) or $\sigma(-\llbracket \mathbf{w}_i, \mathbf{c}_j, \mathbf{t}_k, \mathbf{s}_l \rrbracket)$ (HOSGNS) with their corresponding embedding vectors (negative sampling).

obtained as the product of marginal distributions $P_{\mathcal{D}}(i), P_{\mathcal{D}}(j), P_{\mathcal{D}}(k) \dots$ Computing exactly the objective function in Eq. (3.3) (or the 4th-order analogous) is computationally expensive, but it can be approximated by a sampling strategy: picking positive tuples according to the data distribution $P_{\mathcal{D}}$ and negative ones according to independent sampling $P_{\mathcal{N}}$, HOSGNS objective can be asymptotically approximated through the optimization of the following weighted cross entropy loss:

$$\mathcal{L}^{(\text{bce})} = -\frac{1}{B} \Bigg[ \sum_{(ijk\dots) \sim P_{\mathcal{D}}}^{B} \log \sigma\big(\llbracket \mathbf{w}_i, \mathbf{c}_j, \mathbf{t}_k \dots \rrbracket\big) + \kappa \cdot \sum_{(ijk\dots) \sim P_{\mathcal{N}}}^{B} \log \sigma\big(-\llbracket \mathbf{w}_i, \mathbf{c}_j, \mathbf{t}_k \dots \rrbracket\big) \Bigg] \quad (3.8)$$

where $B$ is the number of the samples drawn in a training step and $\kappa$ is the negative sampling constant. We additionally apply the *warm-up* steps explained in Supplementary Information to speed-up the training convergence.

## 4 EXPERIMENTS

For the experiments we use time-varying graphs collected by the SocioPatterns collaboration (http://www.sociopatterns.org) using wearable proximity sensors that sense the face-to-face proximity relations of individuals wearing them. After training the proposed models (HOSGNS applied to $\mathcal{P}^{(stat)}$, $\mathcal{P}^{(dyn)}$ or $\mathcal{P}^{(stat|dyn)}$) on each dataset, embedding matrices $\mathbf{W}, \mathbf{C}, \mathbf{T}$ (and $\mathbf{S}$ in case of $\mathcal{P}^{(stat)}$) are mapped to embedding vectors $\mathbf{w}_i, \mathbf{c}_j, \mathbf{t}_k$ (and $\mathbf{s}_l$) where $i, j \in \mathcal{V}$ and $k, l \in \mathcal{T}$, and we use them to solve different downstream tasks: *node classification* and *temporal event reconstruction*.

### 4.1 EXPERIMENTAL SETUP

**Datasets.** We performed experiments with both empirical and synthetic datasets describing face-to-face proximity of individuals. We used publicly available empirical contact data collected by the SocioPatterns collaboration (Cattuto et al., 2010), with a temporal resolution of 20 seconds, in a variety of contexts: in a school ("LYONSCHOOL"), a conference ("SFHH"), a hospital ("LH10"), a highschool ("THIERS13"), and in offices ("INVS15") (Génois & Barrat, 2018). This is currently the largest collection of open datasets sensing proximity in the same range and temporal resolution used by modern contact tracing systems. In addition, we used social interactions data generated by the agent-based-model OpenABM-Covid19 (Hinch et al., 2020) to simulate an outbreak of COVID-19 in a urban setting.

We built a time-varying graph from each dataset, and for the empirical data we performed aggregation on 600 seconds time windows, neglecting those snapshots without registered interactions at that time scale. The weight of the link $(i, j, k)$ is the number of events recorded between nodes $(i, j)$ in a certain aggregated window $k$. For synthetic data we maintained the original temporal resolution and we set links weights to 1. Table 1 shows statistics for each dataset.

Table 1: Summary statistics about empirical and synthetic time-varying graph data. In order: number of single nodes $|\mathcal{V}|$, number of steps $|\mathcal{T}|$, number of events $|\mathcal{E}|$, number of active nodes $|\mathcal{V}^{(\mathcal{T})}|$, average weight of events $\frac{1}{|\mathcal{E}|} \sum_{e \in \mathcal{E}} \omega(e)$, nodes density $\frac{|\mathcal{V}^{(\mathcal{T})}|}{|\mathcal{V}||\mathcal{T}|}$ and links density $\frac{2|\mathcal{E}|}{|\mathcal{V}|(|\mathcal{V}|-1)|\mathcal{T}|}$.

|  | Dataset | $\|\mathcal{V}\|$ | $\|\mathcal{T}\|$ | $\|\mathcal{E}\|$ | $\|\mathcal{V}^{(\mathcal{T})}\|$ | Average Weight | Nodes Density | Links Density |
|---|---|---|---|---|---|---|---|---|
| Empirical graphs | LYONSCHOOL | 242 | 104 | 44820 | 17174 | 2.806 | 0.6824 | 0.0148 |
|  | SFHH | 403 | 127 | 17223 | 10815 | 4.079 | 0.2113 | 0.0017 |
|  | LH10 | 76 | 321 | 7435 | 4880 | 4.448 | 0.2000 | 0.0081 |
|  | THIERS13 | 327 | 246 | 35862 | 32546 | 5.256 | 0.4046 | 0.0027 |
|  | INVS15 | 217 | 691 | 18791 | 22451 | 4.164 | 0.1497 | 0.0012 |
| Synthetic graphs | OPENABM-COVID19-2k-100 | 2000 | 100 | 1243551 | 198537 | 1.0 | 0.9927 | 0.0062 |
|  | OPENABM-COVID19-5k-20 | 5000 | 20 | 632523 | 99966 | 1.0 | 0.9997 | 0.0025 |

**Baselines.** We compare our approach with several baseline methods from the literature of time-varying graph embeddings, which learn time-stamped node representations: (1) DYANE (Sato et al., 2019), which learns temporal node embeddings with DEEPWALK, mapping a time-varying graph into a supra-adjacency representation; (2) DYNGEM (Goyal et al., 2017), a deep autoencoder architecture which dynamically reconstructs each graph snapshot initializing model weights with parameters learned in previous time frames; (3) DYNAMICTRIAD (Zhou et al., 2018), which captures structural information and temporal patterns of nodes, modeling the *triadic closure* process. Details about hyper-parameters used in each method can be found in the Supplementary Information.

## 4.2 DOWNSTREAM TASKS

**Node Classification.** The aim of this task is to classify nodes in epidemic states according to a SIR epidemic process with infection rate $\beta$ and recovery rate $\mu$. We simulated 30 realizations of the SIR process on top of each empirical graph with different combinations of parameters $(\beta, \mu)$. We used similar combinations of epidemic parameters and the same dynamical process to produce SIR states as described in Sato et al. (2019). Then we set a logistic regression to classify epidemic states S-I-R assigned to each active node $i^{(k)}$ during the unfolding of the spreading process. We combine the embedding vectors of HOSGNS using the Hadamard (element-wise) product $\mathbf{w}_i \circ \mathbf{t}_k$. We compared with dynamic node embeddings learned from baselines. For fair comparison, all models produce time-stamped node representations with dimension $d = 128$ as input to the logistic regression.

**Temporal Event Reconstruction.** In this task, we aim to determine if an event $(i, j, k)$ is in $\mathcal{H} = (\mathcal{V}, \mathcal{E}, \mathcal{T})$, i.e., if there is an edge between nodes $i$ and $j$ at time $k$. We create a random time-varying graph $\mathcal{H}^* = (\mathcal{V}, \mathcal{E}^*, \mathcal{T})$ with same active nodes $\mathcal{V}^{(\mathcal{T})}$ and a number of $|\mathcal{E}|$ events that are not part of $\mathcal{E}$. Embedding representations learned from $\mathcal{H}$ are used as features to train a logistic regression to predict if a given event $(i, j, k)$ is in $\mathcal{E}$ or in $\mathcal{E}^*$. We combine the embedding vectors of HOSGNS as follows: for HOSGNS$^{(stat)}$, we use the Hadamard product $\mathbf{w}_i \circ \mathbf{c}_j \circ \mathbf{t}_k$; for HOSGNS$^{(dyn)}$ and HOSGNS$^{(stat|dyn)}$, we use $\mathbf{w}_i \circ \mathbf{c}_j \circ \mathbf{t}_k \circ \mathbf{s}_k$. For baseline methods, we aggregate vector embeddings to obtain link-level representations with binary operators (*Average*, *Hadamard*, *Weighted-L1*, *Weighted-L2* and *Concat*) as already used in previous works (Grover & Leskovec, 2016; Tsitsulin et al., 2018). For fair comparison, all models are required produce event representations with dimension $d = 192$.

Tasks were evaluated using train-test split. To avoid information leakage from training to test, we randomly split $\mathcal{V}$ and $\mathcal{T}$ in train and test sets $(\mathcal{V}_{tr}, \mathcal{V}_{ts})$ and $(\mathcal{T}_{tr}, \mathcal{T}_{ts})$, with proportion $70\% - 30\%$. For node classification, only nodes in $\mathcal{V}_{tr}$ at times in $\mathcal{T}_{tr}$ were included in the train set, and only nodes in $\mathcal{V}_{ts}$ at times in $\mathcal{T}_{ts}$ were included in the test set. For temporal event reconstruction, only events with $i, j \in \mathcal{V}_{tr}$ and $k \in \mathcal{T}_{tr}$ were included in the train set, and only events with $i, j \in \mathcal{V}_{ts}$ and $k \in \mathcal{T}_{ts}$ were included in the test set.

### 4.3 RESULTS

In this section we first show downstream task performance results for the empirical datasets, leaving results for synthetic datasets to Supplementary Information. Synthetic datasets are used here to compare the performance of the different approaches in terms of training complexity, by measuring the number of trainable parameters and the training time with fixed number of training steps.

All approaches were evaluated for both downstream tasks in terms of Macro-F1 scores in all datasets. 5 different runs of the embedding model are evaluated on 30 different train-test splits for both downstream tasks. We report the average score with standard error over all splits. In node classification, every SIR realization is assigned to a single embedding run to compute prediction scores. In event reconstruction, a different random realization $\mathcal{H}^*$ is assigned to each train-test subset.

Results for the classification of nodes in epidemic states are shown in Table 2. We report here a subset of $(\beta, \mu)$ but other combinations are available on the Supplementary Information. DYNGEM and DYNAMICTRIAD have low scores, since they are not devised to learn from graph dynamics. HOSGNS$^{(stat)}$ is not able to capture the graph dynamics due to the static nature of $\mathcal{P}^{(stat)}$. DYANE, HOSGNS$^{(stat|dyn)}$ and HOSGNS$^{(dyn)}$ show good performance, with these two HOSGNS variants outperforming DYANE in most of the combinations of datasets and SIR parameters.

Table 2: Macro-F1 scores for classification of nodes in epidemic states according to different SIR epidemic processes over empirical datasets. For each $(\beta, \mu)$ we highlight the two highest scores and underline the best one.

| $(\beta, \mu)$ | Model | Dataset | | | | |
| | | LYONSCHOOL | SFHH | LH10 | THIERS13 | INVS15 |
|---|---|---|---|---|---|---|
| | DYANE | **78.1 ± 0.5** | 67.0 ± 1.2 | 52.5 ± 1.7 | **71.9 ± 0.6** | **64.3 ± 0.8** |
| | DYNGEM | 58.7 ± 2.8 | 35.9 ± 1.1 | 34.5 ± 0.7 | 35.5 ± 1.2 | 58.8 ± 1.1 |
| $(0.25, 0.002)$ | DYNAMICTRIAD | 31.0 ± 0.4 | 28.8 ± 0.4 | 29.9 ± 0.3 | 30.3 ± 0.2 | 30.4 ± 0.2 |
| | HOSGNS$^{(stat)}$ | 55.5 ± 0.8 | 57.3 ± 1.1 | 45.9 ± 0.9 | 46.9 ± 0.7 | 44.5 ± 0.7 |
| | HOSGNS$^{(dyn)}$ | **79.2 ± 0.5** | **69.1 ± 1.1** | **59.6 ± 1.5** | 71.8 ± 1.2 | **64.6 ± 0.7** |
| | HOSGNS$^{(stat|dyn)}$ | 77.4 ± 0.6 | 67.4 ± 1.2 | 59.7 ± 1.2 | **72.5 ± 0.7** | 64.2 ± 1.0 |
| | DYANE | **75.3 ± 0.4** | **71.6 ± 1.9** | 59.0 ± 1.8 | 72.4 ± 0.3 | 65.8 ± 0.6 |
| | DYNGEM | 58.9 ± 2.9 | 37.0 ± 4.1 | 41.0 ± 1.4 | 32.5 ± 1.2 | 59.0 ± 1.2 |
| $(0.125, 0.001)$ | DYNAMICTRIAD | 31.2 ± 0.5 | 35.0 ± 3.3 | 30.5 ± 0.7 | 27.4 ± 0.3 | 29.5 ± 0.2 |
| | HOSGNS$^{(stat)}$ | 56.8 ± 0.9 | 61.8 ± 2.4 | 49.1 ± 1.9 | 47.3 ± 0.6 | 45.9 ± 0.7 |
| | HOSGNS$^{(dyn)}$ | **76.0 ± 0.4** | **71.5 ± 2.0** | 59.6 ± 2.0 | **74.2 ± 0.4** | 65.9 ± 0.6 |
| | HOSGNS$^{(stat|dyn)}$ | 74.6 ± 0.4 | 70.2 ± 1.9 | **59.9 ± 2.3** | **74.8 ± 0.4** | **66.0 ± 0.6** |
| | DYANE | 72.2 ± 0.6 | 64.9 ± 1.7 | **59.0 ± 1.2** | 68.0 ± 0.5 | **60.2 ± 0.5** |
| | DYNGEM | 56.4 ± 2.7 | 35.9 ± 4.1 | 35.8 ± 1.2 | 32.9 ± 1.2 | 55.0 ± 0.6 |
| $(0.0625, 0.002)$ | DYNAMICTRIAD | 29.5 ± 0.5 | 33.1 ± 2.5 | 29.6 ± 0.4 | 27.4 ± 0.3 | 28.4 ± 0.2 |
| | HOSGNS$^{(stat)}$ | 55.5 ± 0.7 | 57.6 ± 2.2 | 49.4 ± 0.8 | 45.5 ± 0.4 | 43.6 ± 0.5 |
| | HOSGNS$^{(dyn)}$ | **73.5 ± 0.5** | 65.7 ± 1.6 | **61.1 ± 1.2** | **69.5 ± 0.3** | 59.6 ± 0.5 |
| | HOSGNS$^{(stat|dyn)}$ | **72.9 ± 0.6** | **66.3 ± 1.9** | 58.2 ± 1.1 | **68.5 ± 0.4** | 59.0 ± 0.7 |

Results for the temporal event reconstruction task are reported in Table 3. Temporal event reconstruction is not performed well by DYNGEM. DYNAMICTRIAD has better performance with Weighted-L1 and Weighted-L2 operators, while DYANE has better performance using Hadamard or Weighted-L2. Since Hadamard product is explicitly used in Eq. (3.1) to optimize HOSGNS, all HOSGNS variants show best scores with this operator. HOSGNS$^{(stat)}$ outperforms all approaches, setting new state-of-the-art results in this task. The $\mathcal{P}^{(dyn)}$ representation used as input to HOSGNS$^{(dyn)}$ does not focus on events but on dynamics, so the performance for event reconstruction is slightly below DYANE, while HOSGNS$^{(stat|dyn)}$ is comparable to DYANE. Results for HOSGNS models using other operators are available in the Supplementary Information. We observe an overall good performance of HOSGNS$^{(stat|dyn)}$ in both downstream tasks, being in almost all cases the second highest score, compared to the other two variants which excel in one task but have lower performance in the other.

**Training Complexity.** We report in Table 4 the number of trainable parameters and training time duration for each considered algorithm, when applied to an empirical graph (LYONSCHOOL) and to the synthetic ones. The proposed HOSGNS model requires a number of trainable parameters that is orders of magnitude smaller than other approaches, with a training time considerably shorter as the number of nodes increases, given a fixed number of training iterations. For this analysis, HOSGNS

Table 3: Macro-F1 scores for temporal event reconstruction in empirical datasets. We highlight in bold the two best scores for each dataset. For baseline models we underline their highest score.

| Model | Operator | LYONSCHOOL | SFHH | Dataset LH10 | THIERS13 | INVS15 |
|---|---|---|---|---|---|---|
| DYANE | Average | $56.4 \pm 0.4$ | $52.9 \pm 0.5$ | $52.3 \pm 0.6$ | $51.0 \pm 0.4$ | $52.7 \pm 0.4$ |
| | Hadamard | $89.7 \pm 0.3$ | $\underline{86.5} \pm 0.3$ | $\underline{74.6} \pm 0.6$ | $94.7 \pm 0.1$ | $94.1 \pm 0.1$ |
| | Weighted-L1 | $90.2 \pm 0.2$ | $83.3 \pm 0.5$ | $73.3 \pm 0.7$ | $94.7 \pm 0.1$ | $94.4 \pm 0.2$ |
| | Weighted-L2 | $\underline{90.6} \pm 0.2$ | $84.5 \pm 0.5$ | $72.0 \pm 0.5$ | $\underline{95.0} \pm 0.1$ | $\underline{94.8} \pm 0.2$ |
| | Concat | $65.7 \pm 0.4$ | $53.8 \pm 0.4$ | $56.2 \pm 0.6$ | $57.0 \pm 0.4$ | $50.9 \pm 0.4$ |
| DYNGEM | Average | $57.8 \pm 0.5$ | $56.8 \pm 0.7$ | $\underline{54.8} \pm 1.5$ | $40.4 \pm 1.5$ | $42.8 \pm 0.9$ |
| | Hadamard | $\underline{62.2} \pm 0.4$ | $55.1 \pm 1.0$ | $52.5 \pm 1.6$ | $40.8 \pm 1.5$ | $43.7 \pm 1.0$ |
| | Weighted-L1 | $58.4 \pm 0.6$ | $52.3 \pm 0.7$ | $50.9 \pm 1.2$ | $\underline{41.3} \pm 1.6$ | $44.8 \pm 0.9$ |
| | Weighted-L2 | $53.7 \pm 0.6$ | $47.0 \pm 0.8$ | $47.0 \pm 1.3$ | $39.2 \pm 1.2$ | $43.6 \pm 0.6$ |
| | Concat | $60.4 \pm 0.4$ | $\underline{57.8} \pm 1.3$ | $48.9 \pm 1.7$ | $36.9 \pm 1.3$ | $\underline{45.7} \pm 1.0$ |
| DYNAMICTRIAD | Average | $51.7 \pm 0.2$ | $56.9 \pm 0.4$ | $60.2 \pm 0.6$ | $58.1 \pm 0.2$ | $56.1 \pm 0.3$ |
| | Hadamard | $60.3 \pm 0.3$ | $58.9 \pm 0.4$ | $59.5 \pm 0.5$ | $62.2 \pm 0.3$ | $64.7 \pm 0.3$ |
| | Weighted-L1 | $\underline{79.1} \pm 0.4$ | $72.3 \pm 0.4$ | $75.5 \pm 0.6$ | $70.8 \pm 0.3$ | $78.1 \pm 0.2$ |
| | Weighted-L2 | $77.4 \pm 0.4$ | $\underline{73.4} \pm 0.4$ | $\underline{77.4} \pm 0.5$ | $\underline{72.4} \pm 0.2$ | $\underline{78.9} \pm 0.3$ |
| | Concat | $52.2 \pm 0.2$ | $53.4 \pm 0.3$ | $55.9 \pm 0.7$ | $55.1 \pm 0.2$ | $53.2 \pm 0.3$ |
| HOSGNS$^{(stat)}$ | Hadamard | $\mathbf{98.5 \pm 0.1}$ | $\mathbf{98.8 \pm 0.1}$ | $\mathbf{99.8 \pm 0.1}$ | $\mathbf{99.6 \pm 0.1}$ | $\mathbf{99.1 \pm 0.1}$ |
| HOSGNS$^{(dyn)}$ | Hadamard | $90.3 \pm 0.2$ | $80.9 \pm 0.4$ | $68.1 \pm 0.7$ | $93.5 \pm 0.2$ | $87.2 \pm 0.2$ |
| HOSGNS$^{(stat\|dyn)}$ | Hadamard | $\mathbf{91.8 \pm 0.2}$ | $\mathbf{86.7 \pm 0.4}$ | $73.6 \pm 0.6$ | $94.3 \pm 0.1$ | $89.0 \pm 0.2$ |

sampling was implemented by picking positive and negative examples from a corpus of random walks sampled from a given graph. For HOSGNS$^{(stat)}$ random walks are sampled from the set of temporal snapshots $\{\mathcal{G}^{(k)}\}_{k \in \mathcal{T}}$ with window size $T = 1$, and for HOSGNS$^{(dyn)}$ random walks are sampled from the supra-adjacency graph $\mathcal{G}_{\mathcal{H}}$ with window size $T = 10$. With these sampling strategies, positive examples are drawn from the same probability distributions as defined in Eq. (3.5) and Eq. (3.6).

Table 4: Number of trainable parameters and training time of each time-varying graph representation learning model compared between LYONSCHOOL and synthetic datasets. The embedding dimension is fixed to 128, technical specifications of the computing system and hyper-parameters configuration are reported in Supplementary Information.

| Model | LYONSCHOOL $\|\mathcal{V}\| = 242, \|\mathcal{T}\| = 104$ | | OPENABM-COVID19-2k-100 $\|\mathcal{V}\| = 2000, \|\mathcal{T}\| = 100$ | | OPENABM-COVID19-5k-20 $\|\mathcal{V}\| = 5000, \|\mathcal{T}\| = 20$ | |
|---|---|---|---|---|---|---|
| | Tr. parameters | Tr. time | Tr. parameters | Tr. time | Tr. parameters | Tr. time |
| DYANE | 4,396,544 | 62s | 50,825,472 | 1,014s | 25,591,296 | 448s |
| DYNGEM | 459,270 | 516s | 1,867,428 | 10,765s | 4,270,428 | 23,307s |
| DYNAMICTRIAD | 3,221,632 | 1,131s | 25,600,128 | 17,191s | 12,800,128 | 12,625s |
| HOSGNS$^{(stat)}$ | 75,264 | 316s | 524,800 | 548s | 1,282,560 | 724s |
| HOSGNS$^{(dyn)}$ | 88,576 | 303s | 537,600 | 565s | 1,285,120 | 734s |

We recall that HOSGNS, by learning disentangled representations of nodes and time intervals, uses a number of parameters in the order of $\mathcal{O}(|\mathcal{V}| + |\mathcal{T}|)$, while models that learn node-time representations (such as DYANE) need a number of parameters that is at least $\mathcal{O}(|\mathcal{V}| \cdot |\mathcal{T}|)$. In the Supplementary Information we include plots with two dimensional projections of these embeddings, showing that the embedding matrices of HOSGNS approaches successfully capture both the structure and the dynamics of the time-varying graph.

## 5 CONCLUSIONS

In this paper, we introduce higher-order skip-gram with negative sampling (HOSGNS) for time-varying graph representation learning. We show that this method is able to disentangle the role of nodes and time, with a small fraction of the number of parameters needed by other methods. The embedding representations learned by HOSGNS outperform other methods in the literature and set new state-of-the-art results for predicting the outcome of dynamical processes and for temporal event reconstruction. We show that HOSGNS can be intuitively applied to time-varying graphs, but this methodology can be easily adapted to solve other representation learning problems that involve multi-modal data and multi-layered graph representations.

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
