# OpenReview forum: "Time-varying Graph Representation Learning via Higher-Order Skip-Gram with Negative Sampling"
_ICLR.cc/2021/Conference — Reject_

### Official Review · AnonReviewer2 · 2020-10-21
**Official Blind Review #2**

**Rating:** 5
**Confidence:** 3

**Review:**

In this paper, the authors propose learning node embeddings of time varying graphs. They extend the ideas from Skip Gram Negative Sampling (SGNS) to time varying graphs. They extend the relationship between SGNS and Matrix Factorization to a tensor setting. The key contribution seems to be learning a static embedding for each node and an embedding for a time step. These embeddings are combined to learn a time-aware node embedding. Experiments on multiple datasets show that the proposed method outperforms related benchmarks.

The paper is well written, and easy to follow. My main concern is with the degree of novelty. I think the work is relevant, but am not convinced that the novelty is sufficient to warrant acceptance. It seems a rather straightforward extension of SGNS as well as the idea that the shifter PMI matrix can be factorized, to a tensor. The datasets the authors use are also quite small, so I'm not convinced that the method is scalable. Indeed, scalable tensor factorization is a challenging problem. It's unclear what the authors gain by casting this as tensor factorization, rather than just solving the SGNS problem like in word2vec. Infact I think that's basically what (3.8) does.

additional comments:
- p3: what's \omega(i, j, t_0)?
- sec 3.1 paragraph 2: "obtained by collecting co-occurrences..." : Can the “jump” in T be arbitrary? How are the (i, j, k) tuples constructed specifically?
- from Table 1: These datasets are quite small. Is there a note on scalability? How large can you go?
- From table 3: The results from HOSGNS are significantly better (sometimes 99%) compared to the baselines. Can you provide some intuition as to why this is? what explains the large jump?
- for the task in Table 3, it seems the (stat) version of the model works a lot better, and adding (dyn) actually makes it worse. Does that mean edge detection is hampered by taking time into account? An explanation would be good.

---

> ### Author Response · Authors · 2020-11-24
> **Response to reviewer#2**
>
>
> We would like to thank the reviewer for the insightful comments and careful reading. Here we address the main concerns of the review:
>
> 1) On the degree of novelty.
> We would like to emphasize that the methodology described in the paper has been presented in the framework of time-varying graph representation learning, but it can be reframed in several other domains characterized by multi-modal data and similar datasets with higher-order dependencies (e.g multi-layered and hyper-graphs), potentially with similar impact that the original skip-gram model has had in representation learning for natural language and graph processing [1,2], recommender systems [3], information retrieval [4]. etc. As a parallel, while tensor factorization (TF) can be considered as an extension of matrix factorization, many intuitions and applications are possible only with TF, and is not considered as a minor improvement.
>
> 2) On scalability experiments.
> We have performed further analysis with two large-scale synthetic contact graphs (2k and 5k nodes) to have more insight about the scalability of the many approaches. Following your insightful observation about solving the HOSGNS problem similarly to word2vec, we can improve the scalability of our approach by implementing a direct sampling strategy of random walks, instead of sampling from a precomputed probability tensor (“Training Complexity” paragraph of Section 4.3 in the revised paper). With this improvement the implicit low-rank factorization property still holds, as in the classical SGNS [5]. In section A.11 of the Appendix we added downstream tasks performances for the two synthetic datasets.
>
> 3) In page 3 \omega(i, j, t_0) is the weight of the link (i, j) at time t_0, as defined in the last paragraph of page 2 (basically the adjacency matrix value A_ij^{(k)} ). We added Figure 1 in the revised paper which visually explains the construction of the supra-adjacency graph for a better understanding.
>
> 4) In Section 3.1 we describe the general 3rd-order SGNS framework ignoring which is the process generating co-occurrences (i, j, k) that are passed to the model. In Section 3.2 we go in detail about the sampling process, in particular a 3-way tuple (i, j, k) corresponds to a temporal event e \in \mathcal{E} sampled according to its weight. This is equivalent to performing random walks on snapshot graphs (like node2vec) and keeping pairs of nodes as positive examples within a window size T=1. In this sense the “jump” T is predetermined by the user before running the model. For 4-way tuples we performed random walks on the supra-adjacency graph with window size T=10. Some works have explored the case when T goes to infinity [6], but we have always fixed T to a finite value.
>
> 5) In temporal event reconstruction, we observe that HOSGNS(stat) performs so well for two reasons: (A) the downstream task consists in predicting links that the embedding model has already seen during training (in this sense it is a link reconstruction task, not a link prediction one), then the model seems to better learn the training data structure respect to baselines.
> (B) HOSGNS(dyn) performs worse, despite the time addiction, because it is trained to reconstruct co-occurrences (and links) of the supra-adjacency graph. Actually in this graph, with reference to the definition in page 3, we don’t have a link (i^{(k)}, j^{(k)}) while the event (i, j, k) still exists in the original time-varying graph. So in (dyn)-graph the information about the existence of (i, j, k) is weaker than (stat)-graph. Conversely the structure of (dyn)-graph makes it useful to predict outcomes of dynamical processes because it incorporates information about temporal paths that is not present in (stat)-graph.
>
> We are happy to discuss further on any of these points.
>
> [1] Efficient estimation of word representations in vector space, Mikolov et al. (2013)
>
> [2] node2vec: Scalable feature learning for networks, Grover et al. (2016)
>
> [3] Item2vec: neural item embedding for collaborative filtering, Barkna et al. (2016)
>
> [4] Query expansion with locally-trained word embeddings. Diaz et al. (2016)
>
> [5] Neural word embedding as implicit matrix factorization, Levy and Goldberg (2014)
>
> [6] InfiniteWalk: Deep Network Embeddings as Laplacian Embeddings with a Nonlinearity. Chanpuriya et al. (2020).

---

### Official Review · AnonReviewer3 · 2020-10-27
**Official Blind Review #3**

**Rating:** 5
**Confidence:** 4

**Review:**

Main Idea

In this paper, the authors studied the problem of time-varying graph embedding problems. The authors generalized skip-gram based graph embedding method to time-varying graphs. The authors show that the method can be used to factorize time-varying graphs as high-order tensors via negative sampling. The authors carried out experiments on several time-resolved proximity networks with comparison to several state-of-art baselines.

Strength
The paper is well written and technically sound. The authors provided theoretical analysis for the approximation of negative sampling to tensor factorization.
The authors carried out extensive experiments on several real-world networks with comparison to strong baselines. The application of SIR node classification task is very interesting.

Weakness
The proposed method does not utilize the special property of time-varying graphs. In the equation (3.1), the positive instances are just single edges while the negative are just for independent marginals. It is not clear how random walk is involved in this setting.
The proposed method is more like an acceleration to traditional tensor factorization method for time-varying graphs. In this case, the authors should (1) include tensor factorization baselines to compare the accuracy and (2) carry out experiments on efficiency comparison with classic tensor factorization methods.
The scalability of the proposed method is another concern. The largest network the authors experiment with has ~200 nodes. It will be good to see scalability experiments for example on synthetic networks.
For the evaluation of temporal event reconstruction. It would be better to use time to separate train/test. Also, it would be better to include some simple static graph embedding baseliness to operate on the network with all time-step graphs combined.

Detailed comments:
Line below equation (2.1), please provide definition for vol(G).

---

> ### Author Response · Authors · 2020-11-24
> **Response to reviewer#3**
>
> We would like to thank the reviewer for the insightful comments and careful reading. Here we address the main concerns of the review:
>
> 1) On scalability experiments.
> We have performed further analysis with two large-scale synthetic contact graphs (2k and 5k nodes) to have more insight about the scalability of the many approaches. We can improve the scalability of our approach by implementing a direct sampling strategy of random walks, instead of sampling from a precomputed probability tensor (“Training Complexity” paragraph of Section 4.3 in the revised paper). With this improvement the implicit low-rank factorization property still holds, as in the classical SGNS [1]. In section A.11 of the Appendix we added downstream tasks performances for the two synthetic datasets.
>
> 2) On the role of random walks in the 3-order setting.
> We use as positive instances temporal edges sampling each tuple (i, j, k) according to the corresponding edge weight. This is equivalent to make random walk realizations on the set of snapshot graphs with a window size T=1. We exploit this equivalence in scalability experiments (Table 4 and “Training complexity” paragraph).
>
> 3) On the sentence “The proposed method is more like an acceleration to traditional tensor factorization method for time-varying graphs”.
> The connections between skip-gram embeddings and traditional low-rank approximation methods is still a matter of intense discussion among researchers, and it goes beyond the theoretical discussion that is present in our paper. In particular a very recent work accepted to Neurips 2020 [2] shows that Logistic PCA-based methods are more powerful to perform graph representation learning respect to traditional SVD approaches, and are more similar to methods smoothed with the logistic function and optimized via back-propagation, which is our case. It is reasonable to suppose that such conclusions also apply when we move from static graphs to the time-varying graph setting. Nevertheless, we are including cp-decomposition baselines in the camera-ready paper.
>
> 4) On the evaluation of temporal event reconstruction.
> Although we do not expect significant changes in performance results, we will add the time-related splitting of train/test sets in the camera-ready paper.
>
> 5) Static graph embedding baselines.
> In section A.9 of the Appendix we included an ablation study of node and time features, using DeepWalk [3] as a baseline for the first ones. Compared with HOSGNS, DeepWalk as expected is clearly inferior in both node classification and link reconstruction tasks.
>
> 6) We have provided definition of vol(G) below equation (2.1).
>
> We are happy to discuss further on any of these points.
>
> [1] Neural word embedding as implicit matrix factorization, Levy and Goldberg (2014)
>
> [2] Node Embeddings and Exact Low-Rank Representations of Complex Networks, Chanpuriya et al.  (2020).
>
> [3] Deepwalk: Online learning of social representations, Perozzi et al. (2014)

---

### Official Review · AnonReviewer4 · 2020-11-01
**Interesting topic and empirical tasks; useful qualitative evaluation but lacks novelty, adequate comparisons and requires improved presentation**

**Rating:** 4
**Confidence:** 4

**Review:**

Summary:
========
The paper proposes an implicit tensor factorization approach for learning time-varying node representations over dynamic networks.  The core method lifts the well-known skip gram based embedding approach from matrix to higher order tensors to support temporal dimensions. The authors claim that such tensor based treatment allows to disentangle the role of node and time. Negative sampling method ( similar to noise contrastive estimation) is extended the higher order tensor setting and incorporated in the cross entropy objective for training. In the experiments, the authors consider five variants of face-to-face proximity data that contains temporal interactions and focuses on tasks of node classification (predicting outcome of SIR epidemic process) and link prediction (in the form of event reconstruction). The proposed method has been compared against two discrete time graph representation learning model and a recently proposed tensor based method. The authors claim that the provided method shows comparable performance with requirement to train lesser number of parameters. Also, the authors provide qualitative analysis in terms of embedding visualizations and goodness of fit plots.


Strengths:
========
- The paper focuses on an important problem of representation learning for time-varying graphs as several real-world networks consist of interactions between nodes that occur and change over time. The higher-order skip gram based method proposed by the authors promises better parameter complexity compared to other methods while retaining or exceeding previous empirical performance.
- Both the high level tasks of predicting outcome of epidemic process and event reconstruction are useful and important realizations of basic node classification and link prediction tasks respectively in the time-dependent setting.
- The authors show that the method provides significant empirical performance on both tasks compared to non-tensor based methods and comparable performance to recently proposed approach in [4].
- I find the qualitative analysis provided by the authors in the appendix in terms of embedding visualization, execution time, goodness of fit and other ablations to be very insightful

Concerns and Improvements:
=========================
- One of the major concerns I have is the novelty of the overall approach. Both skip gram embedding and negative sampling based training is well-known approach. While the authors claim that their key contribution is to use these techniques for higher order structures than a matrix. However, skip-gram type of methods with negative sampling for 3-order tensor have been well studied in relational learning literature [1] for static case. This has also been extended to 4-order tensor to consider temporal dimension and presented in previous ICLR [2]. Hence, the novelty of the technical contribution is not very clear. One of the novel part seems to be learning different representations of node and time. But I could not see how this is more helpful than previous approaches. The author needs to adequately discuss and analyze this and present the outcome in the scenario when they only learn one of the two representations.
- In addition to comparing with the relational learning literature (static and dynamic) both in terms of empirical performance and methodological difference, the authors also miss comparison with a recently proposed tensor based representation learning method for dynamic networks [3]. The paper needs to compare with all these methods and distinguish the technical differences to exactly discern the value of the presented approach.
- The third major concern is the author’s claim on the requirement to use lesser number of parameters. I do not find enough evidence in the paper to support this claim. The author need to exemplify the difference in the number of parameters for particular case compared to DyANE and other methods (see Table V in [1] for example). Also it is important to test how these lesser number of parameters affect the empirical performance. Finally, the execution times in Table 1 do not seem to show any speedup gained due to these less number of parameters. Can the authors discuss more on these effects of lesser number of parameters in addition to retaining performance of DyANE and also compare this with other methods I mentioned above?
- As a follow-up, most experiments are done on graphs with few hundred nodes while real-world networks contains thousands and even more number of nodes. I am not able to see how this method would scale to such large graphs. Does the improvement in parameter complexity above help to promote scalability? Or Is this a limitation of the proposed approach?
- From my understanding, the authors use the warm-up effect for finding a good initialization of the parameters. However, the gap shown in Figure 1 in Appendix for warm-up vs non warm-up case is concerning as it appears that majority of the performance gain (especially considering only marginal performance increase over DyANE) is achieved due to warm-up steps and less due to the effect of the training via negative sampling.
- Overall the paper reads very dense and needs an improved presentation. Specifically, the concepts explained in Section 2 and 3 can be better presented using figures to explain the details such as cross coupling (see Fig 1 in [4] for an example)
- While the authors present better performance for the proposed approach in the experiment section and also provide qualitative analysis in the appendix, it is also important for the authors to discuss and analyze the reasons behind the performance gain with this method in the text in addition to describing the results from the table.

Minor Points not affecting the score:
===============================

- I find the overall performance increase over DyANE to be marginal and while this is not a major concern in itself that has affected my score for the paper, the authors need to provide more comparisons and distinguish their approach from DyANE with better analysis in terms of parameter complexity  to fully support their claim of better performance.
- Minor type - below equation 3.2, should it be P_N(i,j,k)? It seems i and its corresponding term in RHS is missing.

Given the concerns above, I currently do not find this paper ready for publication. I will be happy to revisit my score based on the author’s response to the concerns raised by me above.

References:
==========
[1] A Review of Relational Machine Learning for Knowledge Graphs, Nickel et. al. 2015

[2] Tensor Decompositions for Temporal Knowledge Base Completion, Lacroix et. al. ICLR 2020

[3] Dynamic Graph Convolutional Networks Using the Tensor M-Product, Malik et. al. 2020

[4] DyANE: Dynamics-aware node embedding for temporal networks, Sato et. al. 2020

---

> ### Author Response · Authors · 2020-11-24
> **Response to reviewer#4**
>
> We would like to thank the reviewer for the insightful comments and careful reading. Here we address the main concerns of the review:
>
> 1) On comparisons with respect to existing models.
> We added in section A.10 of the Appendix additional experiments with a 3-order relational learning model (HOLE [1]) and 4-order CP decomposition. Although this preliminary analysis is performed in one dataset, it shows the effectiveness of HOSGNS model against relational learning/tensor decomposition approaches (both in terms of task performance and training time). We did not find a publicly available implementation of model [2] suggested by the reviewer, but we are looking to add one more baseline from dynamic networks embeddings in the camera-ready paper.
>
> 2) On scalability experiments.
> We have performed further analysis with two large-scale synthetic contact graphs (2k and 5k nodes). We can improve the scalability of our approach by implementing a direct sampling strategy of random walks, instead of sampling from the precomputed probability tensor (“Training Complexity” paragraph of Section 4.3). In section A.11 of the Appendix we added downstream tasks performances for the two synthetic datasets.
>
> 3) On the claim to use a lesser number of parameters.
> Learning different representations of node and time is helpful since it makes possible to split a time-dependent node embedding u(t) into a node embedding u and a time embedding t. This property allows us to learn the same set of embeddings {u(t)} as every dynamic graph embedding model, but with less parameters because we learn separately {u} and {t} with ~(N+T)d parameters. On the other hand, embedding models like DyANE compute different time-indexed instances of a node embedding each time that the node is active in a given snapshot, requiring in the worst case NTd parameters. We also show in Table 4 of the revised paper the number of trainable parameters and training times in three datasets characterized by highly different node sizes, noticing the gain in training complexity due to the small fraction of parameters involved.
>
> 4) On the sentence “..present the outcome in the scenario when they only learn one of the two representations”.
> We added in section A.9 of the Appendix an ablation study where we analyze the performance of node and time embeddings separately in downstream tasks,  in comparison with static embedding DeepWalk (for node embeddings) and Positional Encoding (for time embedding).
>
> 5) On the warm-up effect.
> We updated Figure 1 in Appendix with loss functions normalized by a factor (\kappa +1), where \kappa is the negative sampling parameter. This is the correct normalization needed by the weighted cross entropy loss of Eq. (3.8) which allows comparisons between different \kappa values. In particular now we can see that with a random initialization in all cases the loss values start from the same value L_0 (~0.7) before decreasing. With warm-up we optimize L_0 before training and then with the same number of iterations we reach smaller loss values. This difference is not as big as before (without the correct normalization) and makes clearer the efficacy of negative sampling training regardless of the warm-up.
>
> 6) On the overall quality of the presentation.
> We agree with the reviewer about possible improvements on the presentation. We added a figure in page 3 of the revised paper to better explain the structure of the supra-adjacency representation of a time-varying graph.
> We will put a lot of effort in the camera-ready version to fix residual issues in the exposition.
>
> 7)  Reasons behind the performance gain.
> We ascribe the smaller prediction error of the proposed model to the use of a lower number of parameters (the higher the number of parameters, the higher the tendency to overfit). However, while this assertion is true in node classification (where epidemic node labels come from an external process never directly sensed by the algorithm), it is not valid in event reconstruction where we predict links already observed by the model at training time. To prove the capacity of predicting links without overfitting it would be necessary to set an effective temporal event prediction task removing a fraction of links before training HOSGNS, and not after as we have shown in this paper. We leave the event prediction analysis for future work.
>
> 8) Below Eq. (3.2), P_N(j,k) is correct since when we perform negative sampling for an observation (i,j,k) we sample j_N and k_N randomly, keeping fixed i. The reason behind that is the following: if we pick an observation (i, j, k) according to P(i,j,k), the index i by itself is implicitly drawn according to P(i). So to make a negative tuple is sufficient to fix i ( ~P(i) ) and sample j, k ~ P(j)P(k).
>
> We are happy to discuss further on any of these points.
>
> [1] Holographic embeddings of knowledge graphs, Nickel et al. (2015)
>
> [2] Dynamic Graph Convolutional Networks Using the Tensor M-Product, Malik et al. (2020)

---

### Official Review · AnonReviewer1 · 2020-11-10
**A good paper in terms of both theory and practice.**

**Rating:** 7
**Confidence:** 3

**Review:**

Clarity: The motivations of learning representations for  time dependent proximity graphs generated from contact tracing are well explained. The learning of two disentangled representations to capture topological structure and temporal dynamics information is clearly demonstrated.

Novelty: The paper extends a previous proof by Levy and Goldberg, about how Mikolov’s SGNS is implicitly factorizing a word-context matrix, to higher order tensors . This higher order generalisation applied to time-varying graphs is used to learn embeddings, which are shown to effectively encode graph structure and dynamics.

Impact: The paper proposes an extension of a previous well known proof for higher order tensors, which leads to an embedding technique which is shown to perform well on real world datasets
The code and datasets have been made freely available.

Correctness: Several experiments have been conducted to demonstrate effectiveness of learned embeddings
Theoretical groundwork for the proposed model has been well laid.

---

> ### Author Response · Authors · 2020-11-24
> **Response to reviewer#1**
>
> We would like to thank the reviewer for the careful reading and valuable feedback.

---

### Author Response · Authors · 2020-11-24
**General response to reviewers**


We would like to thank the reviewers for the high quality of their reviews, their insightful comments and suggestions. We have made revisions to the paper and added new experiments to address all the constructive comments. We think that the overall quality of the paper improved a lot thanks to the very constructive suggestions from reviewers. We briefly summarize here the major changes and we address all the other concerns in the individual responses.

(i) We added in the main paper an additional figure (Figure 1) to illustrate the supra-adjacency graph representation.

(ii) We included two large-scale synthetic datasets to make additional analysis on model complexity in the main paper (Table 4 and “Training complexity” paragraph) and downstream tasks performances on such datasets (A.11 of the Appendix).

(iii) We included in the Appendix (A.9) an ablation study to highlight separately the contribution of node and time embeddings in downstream tasks.

(iv) We included in the Appendix (A.10) a concise comparison with relational learning and tensor factorization baseline algorithms.

---

### Decision · Program_Chairs · 2021-01-07
**Final Decision**

**Decision:**

Reject

**Comment:**

The paper is concerned with learning representations for time-varying graphs which is an important problem that is relevant to the ICLR community. For this purpose the authors propose a new method to extend skip-gram with negative sampling to higher-order tensors with the goal to perform an implicit tensor factorization of time-varying graphs.The proposed approach shows promising experimental improvements compared to previous methods. Reviewers highlighted also the tasks considered in the paper as well as the theoretical and qualitative analysis as further positive aspects.

However, there exist still concerns regarding the current version of the manuscript. In particular, reviewers raised concerns regarding the novelty of the approach (SGNS, its extension to higher-order tensors, as well as the connection to PMI have been studied in the literature). As such the new technical contributions are limited. Reviewers raised also concerns regarding the scalability of the method and its applicability to large graphs. The revised version addresses this concern to some extent by showing experiments on mid-sized graphs with 2000/5000 nodes. While this clearly improves the paper, I agree with the majority of the reviewers that the manuscript requires an additional revision to iron out the points raised in this round of reviews. However, the presented results are indeed promising and I'd encourage the authors to revise and resubmit their work considering the reviewers' feedback.